# Controlling Large Language Model Agents with Entropic Activation Steering

## Abstract

The rise of large language models (LLMs) has prompted increasing interest in their use as in-context learning agents. At the core of agentic behavior is the capacity for *exploration*, or the ability to actively gather information about the environment. But how do LLM agents explore, and how can we control their exploratory behaviors? To answer these questions, we take a representation-level perspective, and introduce Entropic Activation Steering (EAST), an activation steering method for in-context LLM agents. Firstly, we demonstrate that EAST can effectively manipulate an LLM agent's exploration by directly affecting the high-level actions parsed from the outputs of the LLM, in contrast to token-level temperature sampling. Secondly, we reveal how applying this control modulates the uncertainty exhibited in the LLM's thoughts, guiding the agent towards more exploratory actions. Finally, we demonstrate that the steering vectors obtained by EAST generalize across task variants. In total, these results show that LLM agents explicitly encode uncertainty over their actions in their representation space. Our work paves the way for a new understanding of the functioning of LLM agents and to effective control of their decision-making behaviors.

## 1 Introduction

Successful agentic behavior requires a decision-maker to consider its beliefs about the world while determining which action to take: Should I exploit what I know about the task? Should I search for more information? Can I be sure that my decisions are correct? To build agents that are both effective and reliable, it is paramount to assess whether they are able to autonomously ask these questions, to find answers to them, and to incorporate these answers into their decision-making process.

These considerations are especially important when developing agents built on top of large language models (LLMs). Due to their natural language interface and wide range of capabilities, LLMs hold the promise of powering a new generation of agentic systems. In particular, they have been noted for their ability to perform in-context learning, or the adaptation of their predictions based on examples provided in the prompt. This capability sets the stage for deploying LLMs as in-context learning agents, capable of perceiving the world, executing actions, and achieving diverse human-specified goals by dynamically adapting their behavior in response to feedback from the environment.

However, in contrast to well-studied decision-making algorithms based on reinforcement learning (Sutton & Barto, 2018), relatively little is known about how LLM agents come to their decisions through interaction. While the LLM operates at the token level, playing the role of the reasoning engine behind the agent, decisions happen at a higher level of abstraction, after the output text produced by the LLM is parsed into an action. Overall, the interaction between these two levels is not well understood, and it plays a vital role in determining how the agent's beliefs shape its action distribution.

Indeed, recent work has shown that this process frequently goes awry, causing in-context LLM agents to fail to produce sensible exploratory behavior (Krishnamurthy et al., 2024). They tend to be overconfident, rapidly reducing the uncertainty about their decisions and committing to a particular solution, even when it should be clear that more information is needed. How can we effectively intervene on this behavior?

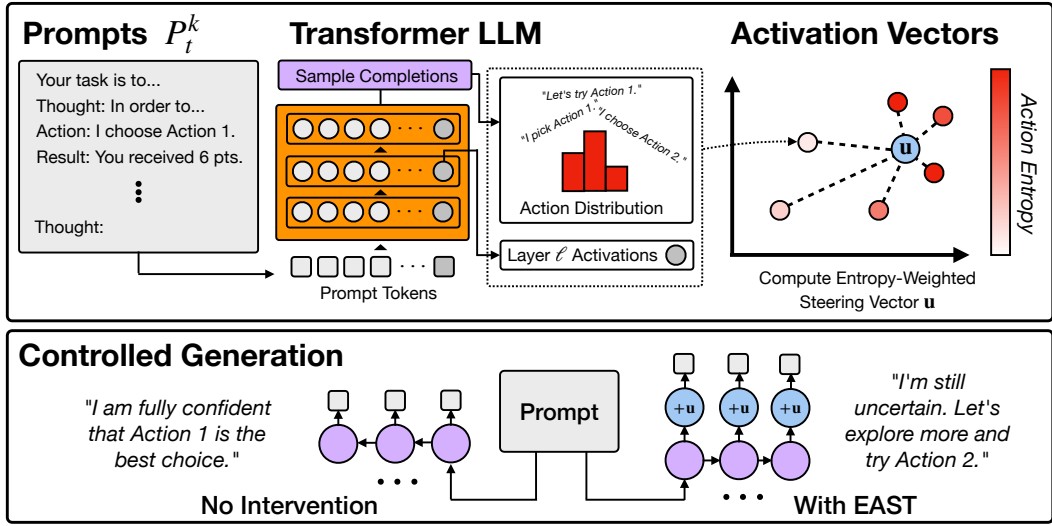

Figure 1: Overview of Entropic Activation Steering (EAST). In Phase 1, the method constructs a steering vector by averaging the activations produced by the LLM agent given a set of prompts, weighting them by the entropy of the resulting action distribution. In Phase 2, during new runs of interactions with the environment, it steers the agent by adding this vector to the LLM's activations at a target layer for each generated token position. The method increases the agent's subjective uncertainty about what to do and leads to more exploratory behavior.

In this paper, we introduce Entropic Activation Steering (EAST), a method to alter an LLM agent's subjective uncertainty over decisions and entropy over actions. EAST uses a dataset of logged interactions between the LLM agent and an environment to obtain a *steering vector*. This vector is computed as an entropy-weighted average of the (run-centered) representations that an LLM produces right before making a decision. Similarly to previous work in activation addition (Rimsky et al., 2023), the steering vector is applied at decision time by adding it, at a specific layer, to the representation corresponding to the tokens that are being generated by the LLM.

EAST directly controls the entropy of the agent's distribution over actions, well beyond what is achievable by simply modifying an LLM's token sampling temperature. Moreover, EAST modifies the subjective uncertainty expressed by an LLM agent in its ReAct-style thoughts (Yao et al., 2022), towards a less exploitative and more information-seeking attitude. With controlled experiments in bandit tasks expressed in language, we show that EAST is able to steer the agent towards more explorative behavior, effectively addressing the overconfidence exhibited by LLM agents.

We demonstrate that EAST generalizes to variations in prompts and LLMs. Surprisingly, we show that the steering vectors we construct can transfer between tasks which are presented as different natural language scenarios, but are equivalent from the sequential decision-making standpoint. Overall, the effectiveness of EAST and our in-depth analyses suggest that LLMs possess an abstract representation of the uncertainty over their decisions, and that it is possible to exercise direct control on it, paving the way to more interpretable and controllable LLM agents.

## 2 BACKGROUND

Modern language models interact with text input through a process of tokenization, in which a body of text is broken down into small units known as tokens (Mielke et al., 2021). To begin, let $\Omega$ be a finite set of natural language tokens. We consider the set of token sequences of finite length, $\Omega^*$, consisting of elements $\omega = (\omega_1, \ldots, \omega_n)$ where $n$ is the length of that sequence.

An LLM is a deep neural network, $f_\theta$, which maps a given sequence of tokens to a categorical distribution over the next token that would follow, which we denote by $p_\theta(\cdot \mid \omega)$. LLMs implement their computations as a sequence of stacked layers, with the network producing intermediate activations corresponding to each input token, $z = f_\theta^\ell(\omega) \in \mathbb{R}^{n \times d}$ for some layer $\ell$ and hidden dimension $d$. We write $z_i \in \mathbb{R}^d$ for the activation corresponding to the $i$-th token.

We are most commonly interested in producing completions $C$ from the model given some prompt $P \in \Omega^*$. This process proceeds by autoregressive sampling. We first sample a token $c_1 \sim p_\theta(\cdot \mid P)$, and then continue by the recurrence relation $c_{k+1} \sim p_\theta(\cdot \mid P, c_1, \ldots, c_k)$, repeating this process until the model generates a special $[\text{EOS}]$ token, yielding the completion $C = (c_1, c_2, \ldots)$. We denote the distribution over completions implied by this process as $\text{LLM}(\cdot \mid P)$.

An *in-context LLM agent* interacts with an environment to perform a task described in an initial prompt $P_0$. At each timestep $t \in \{1, \ldots, T\}$, the model generates a completion $C_t \sim \text{LLM}(\cdot|P_t)$. An action $a_t$ is then extracted by a *parsing function*, mapping the set $\mathcal{C}$ of possible completions to the set $\mathcal{A}$ of possible actions in that environment. We consider this process of completion generation and parsing to represent the agent's stochastic policy over actions given some prompt, which we denote $\pi(\cdot|P_t)$. Note that the model's completions may not always correspond to a valid action. In such cases, the interaction immediately terminates. Once the action $a_t$ is executed in the environment, it returns some text *feedback* $F_t$ to the agent. The interaction is iterated by concatenating the information into a new prompt $P_{t+1} = (P_t, C_t, F_t)$ up to the horizon $T$.

Our experiments focus on a Gaussian multi-armed bandit setting (Lattimore & Szepesvari, 2017), in which the action space $\mathcal{A}$ is a set of possible arm choices and the feedback $F_t$ is a string describing a numerical reward drawn from a Gaussian distribution $\mathcal{N}(\mu_a, \sigma_a)$ associated to a particular arm $a \in \mathcal{A}$. At each round, the agent has to choose which arm to pick. The task description $P_0$ tasks the agent with maximizing the sum of the rewards it receives over time. This setting captures the essential elements of self-evaluation and in-context learning across turns of interaction (Shinn et al., 2023), making them easier to analyze.

## 3 RELATED WORK

By studying how LLM agents represent uncertainty and presenting a steering technique specific for agents, our paper connects recent work in LLM agents and in representation engineering (Zou et al., 2023). We will now provide an overview of the most relevant work from these two research communities.

**LLM-based agents.** LLMs have been recently employed for creating agents, leveraging their capabilities such as proposing actions (Wu et al., 2023), generating code (Wang et al., 2024; Ma et al., 2024), or evaluating outcomes (Klissarov et al., 2024; Kwon et al., 2023). In this paper, we focus on in-context LLM agents, which use the ability of LLMs to learn from data in their prompt (Brown et al., 2020) to process a history of interactions with an environment. We employ an LLM agent multi-armed bandit setup (Krishnamurthy et al., 2024; Park et al., 2024; Schubert et al., 2024; Binz & Schulz, 2023). The advantage of this setup resides in its ability to capture, in a more controlled setting, essential aspects of good decision-making. These systems are typically based on repeated interactions with a task, and heavily rely on the in-context learning abilities of existing LLMs (Shinn et al., 2023; Liu et al., 2023; Mirchandani et al., 2023). An important component in our discussion is the relationship between the token generation process and the action extraction process, which is encountered in recent work using reinforcement learning to train LLMs in decision-making tasks (Zhou et al., 2024).

**Representations of LLMs and activation steering.** Our analyses of the representation space of LLM agents and our EAST method are closely related to recently proposed techniques for activation steering (Subramani et al., 2022; Turner et al., 2023; Rimsky et al., 2023; Li et al., 2023; Wu et al., 2024) and, more broadly, to the recent interest in interpreting the activations of LLMs (Zou et al., 2023; Heimersheim & Nanda, 2024). In particular, similarly to (Rimsky et al., 2023), we apply a steering vector during autoregressive unrolling by adding it to the activations at each position of generated tokens. Differently from these methods, the method we will present focuses on a sequential decision-making setting. Furthermore, we intervene on the action entropy of an LLM agent by leveraging a continuous-valued signal instead of the discrete contrastive approach applied in other recent work (Rimsky et al., 2023; Turner et al., 2023). Our work is related to recent efforts on the mechanistic interpretability of agents using reinforcement learning to navigate gridworlds (Mini et al., 2023), or imitating humans to play chess (Karvonen, 2024). We instead focus on in-context LLM agents based on pretrained models, connecting recent analyses of the representation space of LLMs in a supervised in-context learning setting (Hendel et al., 2023) to agentic use cases.

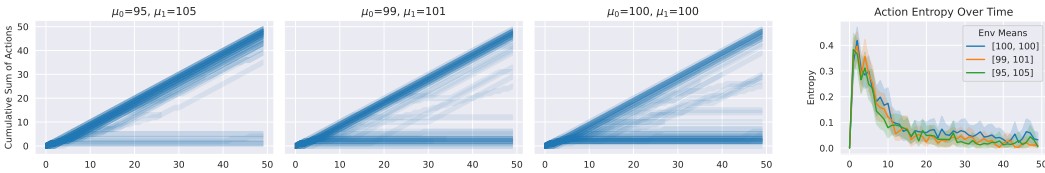

Figure 2: **Left:** Evolution of choices over two actions (0 and 1) taken by LLM agent over time in increasingly ambiguous bandit settings. A darker color corresponds to a more common behavior. The LLM agent tends to commit to a single arm even when choosing should be hard or impossible. **Right:** The evolution of the LLM agent's entropy over actions, over time. The rapid decrease in entropy corresponds to the agent committing to a single action.

## 4   A CLOSER LOOK AT THE UNCERTAINTY OVER ACTIONS OF AN LLM AGENT

**Experimental setting**   Following previous work (Krishnamurthy et al., 2024), we consider two-armed Gaussian bandits with different means $\mu_0, \mu_1$, which we vary depending on the experiment. For ease of analysis, we keep the variances common and fixed to $\sigma_0 = 10, \sigma_1 = 10$ unless otherwise specified. We describe the task to the agent with the prompt in Prompt 1, reported in the appendix (which also reports examples of interactions), in which the two arms are described to the agent as Buttons that it can press. The agent is instructed to evaluate both options in order to maximize its score over time. We use the ReAct prompting (Yao et al., 2022) strategy, which asks the LLM to produce a thought before selecting a particular action. In addition to increasing the reliability of the agent at generating valid actions, inspecting thoughts will also allow us to qualitatively inspect the agent's expression of its subjective uncertainty. For each round of interaction, we generate 25 different completions, parse actions from them, and randomly sample from the valid actions. When estimating the entropy of the action distribution, we consider the set of these valid actions. We study LLMs based on the Transformer architecture (Vaswani et al., 2017). We focus on `Mixtral-8x7B` (Jiang et al., 2024), and also report results for `DBRX` (Databricks, 2024) in Section 6.3. In all cases, the agent-environment interaction is implemented as a dialogue, and we correspondingly use the instruction-tuned versions of these models. Each error bar displayed in the paper shows a bootstrapped 95% confidence interval around the mean, computed using the default behavior of the `seaborn` python library (Waskom, 2021).

### 4.1   THE BEHAVIOR OF IN-CONTEXT LLM AGENTS IN BANDIT TASKS

Previous work (Krishnamurthy et al., 2024) has established that a common failure case of current in-context LLM agents comes from overconfident behavior. In the context of a bandit, this overconfidence corresponds to the agent committing to a particular action without sufficient evidence that that particular action is the best one (i.e. leading to a higher expected reward).

To take a closer look, we plot the evolution of the LLM agent's actions over time, by computing, for each independent run of interaction between the LLM agent and the environment, a cumulative sum over time of the index corresponding to the action selected at time $t$. Thus, for a run of length $T$, a cumulative sum of 0 corresponds to the agent always selecting action 0 and a cumulative sum of $T$ corresponds to the agent selecting action 1.

In Figure 2, we visualize the results of 65 runs of interaction for each of three distinct parameterizations of the environment means, where the standard deviations are fixed at $\sigma_0 = 10, \sigma_1 = 10$. On the plot, each run is represented as a shaded area centered around the line showing this cumulative sum at each timestep. In particular, when the line proceeds horizontally in time, it means the agent selected action 0 at that step, and diagonally, action 1. In aggregate, the opacity of the plot displays the relative frequency of behaviors of the LLM agent, with a darker color corresponding to higher empirical frequency of that behavior. The plot demonstrates that the agent has a strong tendency to commit to a particular action after a small number of steps, represented by horizontal and diagonal shaded areas for actions 0 and 1, respectively. While this behavior could be seen as advantageous in the case where the arms are far apart ($\mu_0 = 95, \mu_1 = 105$), it becomes increasingly irrational as the task becomes harder ($\mu_0 = 99, \mu_1 = 101$), where we observe that the agent commonly commits to the wrong action based on limited data. Even in the extreme case in which both the actions have

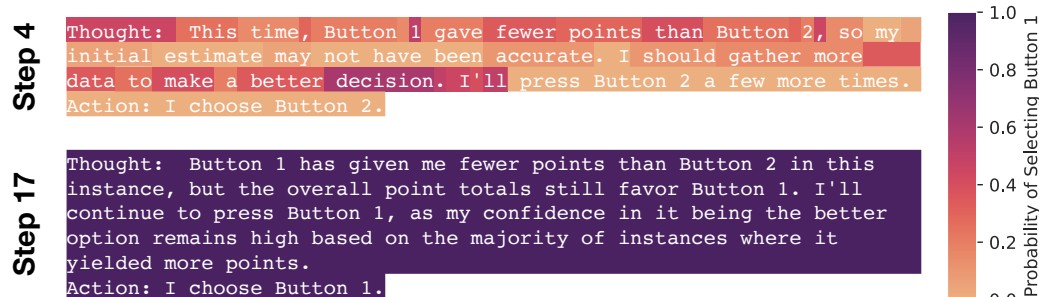

Figure 3: Example of the interaction between token-level sampling and action-level sampling for a two-armed bandit, showing the evolution of the probability that the first action is ultimately selected as the tokens are generated by the LLM.

exactly the same mean ($\mu_0 = 100$, $\mu_1 = 100$), in which we would expect a rational agent to explore indefinitely, the agent still overwhelmingly defaults to arbitrarily selecting one action.

To provide another perspective on this phenomenon, Figure 2 (right) shows the evolution of the entropy of the agent's action distribution $H(\pi(\cdot \mid P_t))$ as time passes, averaged over the different runs. For all the different configurations, the entropy of the LLM agent's action distribution rapidly decreases over time, resulting in insufficient exploration of the available options.

### 4.2 CONNECTING TOKEN AND ACTION GENERATION

Before intervening on the overconfident behavior of the LLM agent, let us dive deeper into the action generation mechanism itself. As described in detail in Section 2, the action generation process relies on the underlying LLM being unrolled to produce a completion (which includes both a thought and a proposed action) and on parsing from this completion an action to be executed in the environment. Thus, each generated token has the potential to contribute to the final decision about the action.

To visualize this process, we show in Figure 3 how token generation and action selection are connected in practice by inspecting the distribution of the agent's actions as its response grows. Following each generated token, we unroll a number $S = 20$ of full generations from the model, parse the resulting action, and estimate the probability of the agent selecting the first action from its empirical frequency across generations. Thus, for each token, we have a corresponding probability of selecting a particular action, which we denote with color in the plot, and we can track this probability throughout of a generation to see how decisions emerge from tokens.

In particular, we observe the evolution of the probability of selecting the first button in two steps far in time (step 4 and step 17) in an example run. While in early steps (see step 4) individual tokens in the LLM's thought progressively determine the action, in later steps (see step 17) the decrease in entropy highlighted in Figure 2 is associated with the evolution of the thought having no effect on the agent's ultimate decision. Echoing previous work that has been done on different forms of chain-of-thought

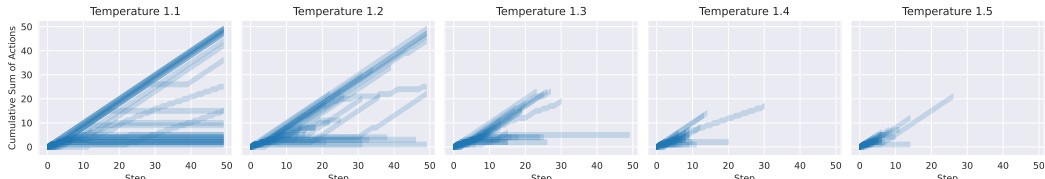

Figure 4: Distribution of choices over two actions (0 and 1) taken by the LLM agent over time when varying the sampling temperature. A darker color corresponds to a more common behavior, and incomplete lines are due to the episode terminating early because of invalid actions. Increasing temperature until the point at which no action can be parsed from the LLM's generations does not significantly change the entropy in action distribution.

prompting (Turpin et al., 2024), the example shows that a model does not necessarily come to a conclusion at the end of the thought, and that the thought acts as a manifestation of an underlying computational process happening in the representation space, but not always as the only guide to a model's final decision.

Having seen the connection between the token-generation and the action-generation processes, it is natural to ask how much intervening on the former can influence the latter, and whether an intervention can counteract the tendency of the LLM agent to be overconfident. The most direct strategy to try to increase the entropy in the generated actions $\pi(\cdot \mid P)$ is to manipulate the entropy in the generated tokens, which is typically achieved by increasing the temperature used during sampling. In Figure 4, we visualize the distribution of agent behaviors on the equal means environment, across runs, for various values of sampling temperature, progressively increasing it up to the point at which the model fails to consistently produce completions from which a valid action can be parsed. The results show that temperature does not significantly change the tendency of the model to overcommit, until no run can be completed. This shows that, due to the nature of their interaction, increasing entropy in token generation does not increase the entropy in the action distribution.

## 5 ENTROPIC ACTIVATION STEERING

In the previous section, we have shown that changing the token sampling temperature does not have a significant effect on the action distribution of the agent. Now, our motivation is to 1) identify at a mechanistic level what drives exploratory behavior for an LLM agent and 2) develop new controls on this behavior.

A natural candidate for doing this is the class of recent methods based on activation steering, which derive *steering vectors* from datasets of LLM representations which are iteratively added to the model's activations during language model generation. These steering vectors are able to modulate complex concepts such as refusal, sycophancy, or hallucination in an LLM's outputs; the success of this intervention also suggests that the steering vector directions represent semantically meaningful concepts in LLM activation space (Rimsky et al., 2023; Arditi et al., 2024; Turner et al., 2023).

However, existing activation steering techniques are insufficient for intervening on the entropy of the action distribution of an LLM agent, for two main reasons.

1. They typically assume access to a dataset with discrete labels for each prompt e.g. "harmful" or "safe". In our setting, we are instead interested in controlling a continuous variable.

2. They are designed for non-agentic settings, in which each prompt is an i.i.d. sample from a distribution and there is no feedback loop of interaction with an external system.

To overcome these shortcomings, we now introduce Entropic Activation Steering (EAST), an activation steering method that directly controls the LLM's action entropy and subjective uncertainty by intervening on its forward pass. EAST consists of two phases: first, computing a steering vector from a dataset of interactions, and second, using the steering vector to modify the behavior of the agent.

In the first phase, given a dataset of prompts $P_t^k$ obtained by letting the agent interact for $K$ runs of $T$ timesteps each, we compute the activations $z_t^k = f^\ell(P_t^k)$ by giving a prompt $P_t^k$ as input, forward-passing the LLM, and extracting the layer-$\ell$ representation corresponding to the last token in the prompt. Then, for each prompt, we estimate the entropy $h_t^k = -\sum_{a \in \mathcal{A}} \pi(a|P_t^k) \log(\pi(a|P_t^k))$ of the action distribution, by generating $M$ different completions from the LLM, extracting the corresponding action, and computing the entropy on the sampled actions. In practice, we use $M = 25$ and only compute the entropy using completions for which the action is successfully parsed. Then, we compute the steering vector as:

$$\boldsymbol{u} = \frac{1}{Z} \sum_{k=1}^{K} \sum_{t=1}^{T} \underbrace{h_t^k}_{\text{Entropy weight}} \left( z_t^k - \underbrace{\frac{1}{T} \sum_{t'=1}^{T} z_{t'}^k}_{\substack{\text{Average activation} \\ \text{in a run}}} \right), \tag{1}$$

with $Z = \sum_{k=1}^{K} \sum_{t=1}^{T} h_t^k$ a normalizing constant. The steering vector is an entropy-weighted average of the activations in the dataset, in which each activation is centered around the mean of the corresponding run's activations.

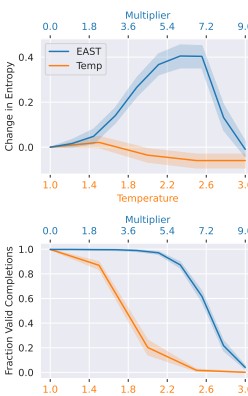

**Model response (no steering)**

Thought: My most recent result for Button 2 is a significant improvement and further justifies my confidence in Button 2. It's now clear that Button 2 has more potential in the long run, so I will continue pressing it.
Action: I choose Button 2.

**Model response (EAST)**

Thought: This time, I received quite a high result for Button 2. It appears that there is still significant variability in Button 2's results, but now it seems that the variability for Button 1 is also high. I'll press Button 1 once more to determine if I should continue with Button 2 or explore further.
Action: I choose Button 1.

Figure 5: Effect of the application of EAST on the LLM agent's actions and thoughts. In contrast to varying the token-level sampling temperature, EAST significantly changes the action entropy for a wide range of multipliers before invalidating a model's completions (left), and affects the agent's subjective uncertainty, steering its thoughts towards more explorative behavior given the same starting situation (right).

The use of an entropy weight $h_t^k$ generalizes the process for steering vector computation proposed in previous work to a continuous setting in the following way. Existing methods typically work in a contrastive fashion, obtaining the steering vector by subtracting the average representation of positive examples from the average representation of negative examples. We can formally see this process as one of defining a weight vector $w$ and then taking an average of the representations in the dataset weighted according to $w$. Through this lens, we can view existing contrastive methods as assigning components $w_i$ such that $w_i \in \{-1, 1\}$ according to the label in the dataset. In this perspective, one can interpret the entropy weight $h_t^k$ as a continuous target, representing an extension of the implicit weighting implied by existing methods.

To handle the interactive nature of the decision task, EAST aggregates over independent runs $P^k$ and timesteps within those runs $P_t^k$. In contrast to the naïve approach of simply summing the corresponding activations, EAST normalizes each activation $z_t^k$ within a run by the average activation over that run. We found in preliminary experiments that this approach is essential to produce functioning steering vectors. This suggests that as the interaction history encoded in the prompt grows, LLM representations specialize to the particular events of that history; our normalization method works to target the specific component of representation space responsible for explorative behavior that is common across runs.

Overall, the first phase extracts a representation whose direction is aligned with the direction that leads, on average, to high entropy. In the second phase, we apply the steering vector to influence the LLM agent's behavior. While generating a completion, we add the steering vector $\boldsymbol{u}$, at each step, to the representation produced by the model at layer $\ell$ at the position of the last token. This yields a steered representation $\hat{z}_i = z_i + \beta \boldsymbol{u}$, where $\beta$ is a multiplier determining the amount of steering. Note that, when generating subsequent tokens after having applied the intervention on a previous activation, we keep that previous activation in the modified state until the action is executed.

## 6 EXPERIMENTS

### 6.1 EAST CAN CONTROL AN LLM AGENT'S UNCERTAINTY OVER ACTIONS

**Experimental setting.** We obtain the steering vector by running EAST on prompts generated in the equal means environment, and evaluate the method on a validation set of 100 prompts $P$ sampled at random from across interactions with differently parameterized environments (see appendix for details). For a given choice of layer $\ell$ and multiplier $\beta$, we measure the average entropy of the model's actions $\pi(\cdot | P)$ across the dataset. When not specified, we use $\ell = 16$ as a layer of the network and a multiplier value of $\beta = 2$.

In Figure 5, we compare EAST's effect on the entropy of the actions produced by the LLM to the one induced by changing temperature during token generation. For a fair comparison, we consider the full ranges of the two relevant hyperparameters, multiplier $\beta$ for EAST and temperature value for temperature-based token sampling, and show the fraction of valid completions generated by each method.

The results show that, by increasing EAST's multiplier, we can significantly increase the entropy in the actions, while variations in temperature have negligible effect on it (note that the maximum attainable entropy in this setting is $\log 2 \approx 0.69$). The same figure shows, on the right, an example of two completions of the model originating from the same prompt, with or without the steering provided by EAST. Not only EAST changes the entropy in the action distribution, but it also induces the model to produce thoughts, for the same situation, that hint at more explorative or uncertain behavior.

We can now analyze the behavior of an agent steered by EAST during its interactions with the environment by going back to the visualization technique employed in Section 4. In Figure 6 we show how EAST affects the distribution of actions produced by the LLM agent during different runs, compared to the agent with no steering applied. The agent steered by EAST is significantly less prone to committing early to a particular arm in the different settings, showing that our method can be used to encourage an LLM agent to explore more in its environment.

We already hinted, with the example in Figure 5, that, in addition to changing the entropy of an LLM agent's action distribution, EAST is also able to steer an agent's verbalized subjective uncertainty as expressed in its thoughts. To have an aggregated visualization of the content of the thoughts of the LLM agent, we gather the top words in terms of relative frequency across different runs of interactions of the LLM agent with the environment, with or without applying EAST (see Section A.1.4 for details). Table 1 shows the top words in the two cases. By default, the thoughts of the LLM model often include terms related to overconfidence and exploitative behavior, such as 'reinforces', 'maximize', or 'superior'. By contrast, applying EAST produces a remarkable qualitative change in the LLM agent's thoughts, which become more related to uncertainty and exploration, with frequent words such as 'variance', 'volatile', or 'uncertainty'.

Taken together, these results demonstrate that, by operating on the representation space of an LLM gent, EAST is able to steer the model away from its overconfident behavior, well beyond what is achievable via sampling temperature, and to manipulate the subjective uncertainty about its decisions. This shows that an LLM possesses and uses an explicit representation of such a concept.

## 6.2 Understanding steering vectors

**Effectiveness of steering vectors at different target layers.** EAST requires a choice of the layer in the LLM that will be used during its two phases, with an impact on both the computation of the steering vector, and on the application of the vector during the interactions of the agent with the environment. We show in Figure 7 that, regardless of the choice for the multiplier $\beta$, the layers at which EAST's intervention is most effective sit in the middle of the LLM, with a peak at the 16th layer, which we used in the rest of our experiments. This is in line with previous work on interpreting the representations of pretrained LLMs outside of the agentic setting, which found that

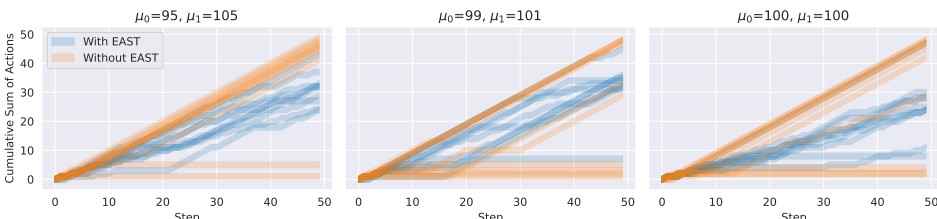

Figure 6: Effect of EAST on the distribution of actions executed in different runs in various bandit problems. EAST's effect on an LLM agent's representation effectively guides the agent towards more explorative behaviors, steering it away from its typical overconfidence.

| No steering | | | EAST | | |
|---|---|---|---|---|---|
| repeatedly | experience | optimize | variance | rounds | uncertainty |
| supports | reaffirms | selecting | moving | comparing | tests |
| superior | maximize | remarkable | maximum | trials | final |
| maintaining | reinforces | historically | feel | dropped | volatility |
| strategy | valid | rewards | volatile | couple | recalculate |
| reason | belief | rewarding | hand | anomaly | starts |

Table 1: Top words in terms of relative frequency present in the thoughts of the LLM agent across different runs, without steering and with steering provided by EAST. EAST modifies an LLM's thoughts towards expressions of subjective uncertainty.

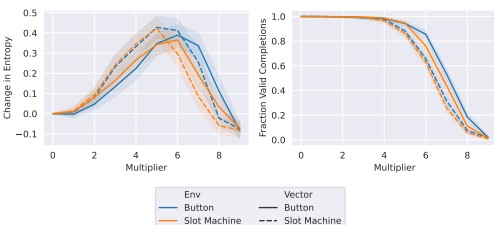 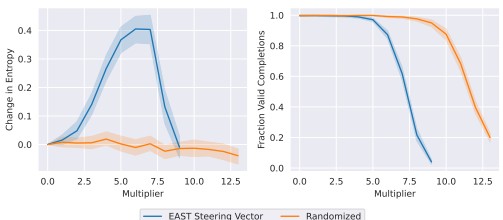

Figure 8: Effect of applying steering vectors derived from two different natural language descriptions of a task to agents prompted with the two descriptions. Steering vectors generated by EAST generalize across task descriptions.

Figure 9: Comparison of effect of EAST's steering vector with a shuffled version of the same vector. EAST's steering effect is due to the direction it is able to find in an LLM's representation space.

the representation of abstract concepts such as sycophancy and refusal resides in layers roughly in the middle of the LLM (Rimsky et al., 2023).

**Importance of the direction of the steering vector.** To solidify the interpretability value provided by EAST, we now give evidence that the increase in action entropy caused by the steering vector is indeed caused by a special direction related to uncertainty in decision-making, as opposed to being simply the effect of any perturbation to an LLM's forward pass. We construct a vector with exactly the same statistics as the steering vector by shuffling its features, and apply this randomized steering vector in the same way we normally do in EAST. Figure 9 shows the result of the comparison with EAST: We find that the randomized vector does not produce any change in the entropy of the action distribution of the LLM agent, highlighting the importance and effectiveness of the direction found by EAST in the representation space of the LLM.

### 6.3 GENERALIZING EAST ACROSS TASKS AND LLMs

**Steering vectors and task description.** We now look at how EAST reacts to differences in the task description provided to the LLM in the initial prompt $P_0$, and try to understand whether

Figure 7: Change in action entropy observed running EAST using different layers. Applying EAST to middle layers is effective, hinting at the fact that the model represents uncertainty over its actions in the middle of the network.

the steering vector captures any concept of uncertainty about the actions that goes beyond a specific prompt. To investigate this, we keep the same problem structure and general description, but switch the entities involved in the sequential decision-making problem from the agent interacting with buttons to playing slot machines (see the appendix for the complete prompt). In particular, we are interested in trying how steering vectors computed in the button and the slot machine settings behave when applied to an LLM agent interacting with either of the two settings. In Figure 8, we show the results of trying all four possible combinations of computation of the steering vector and interaction-time application, in terms of effect on the action entropy of the LLM agent. Strikingly, the results

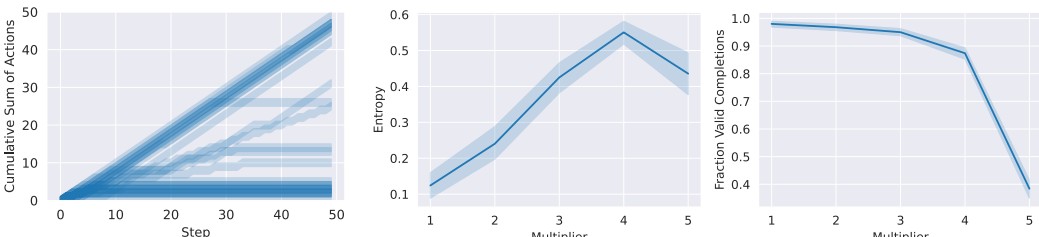

Figure 10: **Left:** Decisions made by `DBRX` over time when interacting with the Buttons task with $\mu_0 = 100, \mu_1 = 100$. Even in this extreme case where one would expect a rational agent to exhibit extended exploration, the model still commits to a single action after a short period of time. **Right:** Results of applying EAST to a validation set of 100 prompts randomly sampled across interactions of `DBRX` with the equal means task. As with `Mixtral-8x7B`, the approach considerably increases the uncertainty in generated actions before significantly affecting the rate of valid completions.

show that not only EAST generalizes across prompt variations, but that steering vectors seamlessly transfer across the different prompt settings. This points at the fact that the LLM agent creates a representation of the uncertainty about its decision-making choices, regardless of the particular entities mentioned in the task description.

**Effectiveness of EAST on other LLMs.** As mentioned at the beginning of the section, we employed a `Mixtral-8x7b` model in most of our experiments, since it provides a good tradeoff between inference speed and performance. To demonstrate the generality of EAST, we conduct additional experiments on the `DBRX` open LLM (Databricks, 2024). We repeat experiments detailed in Section 4.1 and Section 6.1 using this model. The results pictured in Figure 10 show that this model behaves similarly to `Mixral-8x7b`, both in its default strategies on the bandit tasks and its response to the EAST intervention. This demonstrates that, despite the specific information encoded in an LLM's representation depends on its training data and the exact training procedure that was used to train it, EAST can correctly identify the appropriate direction in the representation space and intervene on the LLM agent's behavior.

## 7 CONCLUSIONS

In this paper, we studied how in-context LLM agents behave in sequential decision-making tasks and how they represent uncertainty over actions. After having established that they tend to be overconfident about their decisions, we introduced Entropic Activation Steering (EAST), a method for influencing their exploration. We illustrated how token-level sampling and action generation interact, and demonstrated that EAST can increase the entropy of an LLM agent's action distribution and alleviate its overconfidence, well beyond what is achievable by increasing the sampling temperature at the token level. In addition, we have shown that EAST can modify the subjective uncertainty of an LLM agent, influencing its thoughts towards more uncertain and explorative attitudes.

We believe that EAST can be used as a building block to steer an agent's exploration in future LLM-based systems and that EAST's demonstration that LLMs explicitly represent uncertainty about actions can inform the design of such systems. As designers of agentic LLM-based systems, it is paramount for us to be able to interpret how they make decisions and to steer them towards more desirable behaviors. EAST advances our understanding of the representation that in-context LLM agents have about their uncertainty over decisions, and our ability to control it. Considering that uncertainty over one's actions is a fundamental aspect of successful decision-making, we believe our work to be a promising step in the development of interpretable and steerable in-context LLM agents.

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

# A  APPENDIX

## A.1  ADDITIONAL EXPERIMENTAL DETAILS

### A.1.1  EXPERIMENTAL SETTING

We now describe more details about the experimental setting employed in Section 6, going over how the prompts were generated and outlining the relevant details figure by figure.

We generate datasets of prompts $P_t^k$ by logging the text produced by 65 runs of interaction with the equal means environment. We use horizon $T = 50$, finding that the average run completes more than $98\%$ of those steps.

We evaluate on 100 prompts drawn from random steps of interactions with the four bandits with means $(\mu_0 = 95, \mu_1 = 105)$, $(\mu_0 = 99, \mu_1 = 101)$, $(\mu_0 = 101, \mu_1 = 99)$, $(\mu_0 = 105, \mu_1 = 95)$ for the experiments in Figure 5, Figure 7, and Figure 9, and on means $(\mu_0 = 100, \mu_1 = 100)$ for the experiments from Figure 8. We use 15 completions to estimate the entropy during evaluation.

### A.1.2  LANGUAGE MODEL ASSETS

We conduct experiments on `Mixtral-8x7b` model Jiang et al. (2024), available at this link, and the `DBRX` model Databricks (2024) available here. Mixtral is released under the Apache 2.0 license, and DBRX is released under the Databricks Open Model License.

### A.1.3  COMPUTATIONAL RESOURCES

All experiments were run on an internal compute cluster. All experiments require 8 CPUs and 32GB of memory. Because reproducing the experiments requires a large amount of LLM inference, we will focus the discussion here primarily on the GPU hardware and time used, as this is the main bottleneck.

The computational work required to reproduce the paper breaks down into a few types of experiments. First, running interactions between the LLM and the bandit task: With $T = 50$ steps and $M = 25$ completions per step, each single run requires about 10 minutes on 4x Nvidia A100 80GB GPUs, or 40 minutes in single GPU-minutes. This means that the results in Figure 2 took $3 * 65 * 40$ GPU-minutes = 130 GPU-hours. Extrapolating similarly to the experiments pictured in Figures 4 and the controlled interactions in Figure 6 produces a total estimate of 150 GPU-hours.

The EAST method itself is computationally inexpensive. Given the dataset of prompts $\{P_t^k\}$ we used in Section 6.1 of size 3250, it requires computing the last-token activation for each prompt, a process which takes 1 GPU-hour on the same hardware mentioned above. Then, constructing the steering vector is a near-instant process of computing a weighted average, given the action entropies which were already recorded during the interaction stage.

Finally, computational resources were also dedicated to understanding the effects of EAST on a validation set of 100 prompts, described in Section 6.2. These experiments require sweeping over a large range of layers and multipliers at the cost of 20 GPU-minutes per layer and multiplier. As such, for all experiments in Section 6.2 the cost is approximately 100 GPU hours.

As such, the entirety of experiments in the paper required approximately 260 GPU-hours. The entirety of work for the paper, including preliminary experiments, required an estimated 5x of this figure.

### A.1.4  COMPUTATION OF RELATIVE FREQUENCY

In Table 1, Section 6.1, we report the top words of completions generated by the model during interactions with the task for two conditions, by default and under steering with EAST. We consider all model responses from 10 seeds of interaction per-environment across means $(\mu_0 = 95, \mu_1 = 105)$, $(\mu_0 = 99, \mu_1 = 101)$, and $(\mu_0 = 100, \mu_1 = 100)$ for both conditions.

For each condition, we tokenize all the responses into words. We then remove common English stopwords using the list at `https://github.com/stopwords-iso/stopwords-en`. For each word, we compute its frequency within its respective corpus corresponding to the default and

steered conditions. Then, we compute a relative frequency score for each word as the ratio between the frequencies in the default and steered conditions, and vice versa, to identify the top words.

## A.2 PROMPTS AND EXAMPLE INTERACTIONS

We show here the prompts we used and some notable examples of interactions.

```
[INST] You are an agent playing a game.

You are standing in front of 2 buttons. Each button, when pressed, gives you a random number of points.
Your goal is to accumulate the most points possible by figuring out what button gives more points on
average and pressing it a lot. On each turn, you press one button one time, and I tell you how many
points you got. Make sure to sufficiently explore to make sure you're fully confident which button is
better.

You should format your answer as a Thought, followed by an Action. For example, we might have an
interaction like this:

Let's begin:
Thought: <explain your thinking>
Action: I choose Button <the button you choose, 1 or 2>.

Be sure to think step-by-step in order to identify the button which usually yields the most points so
you can press that one a lot.

Let's begin:
[/INST]
```

Prompt 1: The prompt which we use to describe the bandit task to the agent.

```
[INST] You are an agent playing a game.

You are standing in front of 2 slot machines. Each slot machine, when played, gives you a random number
of dollars. Your goal is to accumulate the most dollars possible by figuring out what slot machine
gives more dollars on average and playing it a lot. On each turn, you play one slot machine one time,
and I tell you how many dollars you got. Make sure to sufficiently explore to make sure you're fully
confident which slot machine is better.

You should format your answer as a Thought, followed by an Action. For example, we might have an
interaction like this:

Let's begin:
Thought: <explain your thinking>
Action: I choose Slot Machine <the slot machine you choose, 1 or 2>.

Be sure to think step-by-step in order to identify the slot machine which usually yields the most
dollars so you can play that one a lot.

Let's begin:
[/INST]
```

Prompt 2: The alternative prompt in which the task is described to the agent as interacting with slot machines rather than buttons that is mentioned in Section 6.2. Other details are kept fixed.

```
[INST] You are an agent playing a game.

You are standing in front of 2 buttons. Each button, when pressed, gives you a random number of points.
Your goal is to accumulate the most points possible by figuring out what button gives more points on
average and pressing it a lot. On each turn, you press one button one time, and I tell you how many
points you got. Make sure to sufficiently explore to make sure you're fully confident which button is
better.

You should format your answer as a Thought, followed by an Action. For example, we might have an
interaction like this:

Let's begin:
Thought: <explain your thinking>
Action: I choose Button <the button you choose, 1 or 2>.

Be sure to think step-by-step in order to identify the button which usually yields the most points so
you can press that one a lot.

Let's begin:
 [/INST]Thought: I need to determine which button is more likely to give me a higher number of points.
 In order to do this, I will press each button a few times to get an initial idea of the average points
 for each button. After that, I will continue to refine my understanding by pressing the button that
 seems to have a higher average, while also occasionally checking the other button to ensure that its
 average hasn't significantly increased.

Action: I choose Button 1.
[INST] Result: You received 101.28 points.
 [/INST]Thought: I received 101.28 points from Button 1. This is a relatively high number, but I should
 still press Button 2 a few times to compare. After exploring both buttons, I will focus on the button
 with the higher average and continue to press it in order to maximize my points.

Action: I choose Button 2.
[INST] Result: You received 84.72 points.
 [/INST]Thought: I received 84.72 points from Button 2, which is less than what I received from Button
 1. Based on this initial exploration, Button 1 seems to be the better option. However, I will continue
 to press Button 2 a few more times to ensure that its average hasn't changed significantly. Then, I
 will focus on pressing Button 1 to maximize my points.

Action: I choose Button 2.
[INST] Result: You received 100.67 points.
 [/INST]Thought: I received a higher number of points from Button 2 this time, but the average is still
 lower compared to Button 1. To ensure that I have an accurate understanding of the averages, I will
 press each button a few more times before focusing on the button that seems to be more profitable.

Action: I choose Button 1.
[INST] Result: You received 94.06 points.
 [/INST]Thought: After this round, Button 1 gave me 94.06 points, which is still higher than the
 average I've seen from Button 2. With these results, I am more confident that Button 1 is the better
 choice. I will now focus on pressing Button 1 frequently to maximize my points.

Action: I choose Button 1.

...

 [/INST]Thought: Wow, I received an impressive 115.54 points from Button 1! This result further
 underscores my confidence in this button's higher average compared to Button 2. I will stay focused on
 pressing Button 1 frequently to maximize my points.

Action: I choose Button 1.
[INST] Result: You received 77.47 points.
 [/INST]Thought: Although Button 1 offered a relatively lower result of 77.47 points, it remains higher
 than the average for Button 2. Additionally, the overall average for Button 1 is still significantly
 better. I will continue to devote my attention to Button 1 and press it frequently to maximize my
 points, with occasional checks on Button 2 to ensure its average hasn't changed drastically.

Action: I choose Button 1.
[INST] Result: You received 108.72 points.
 [/INST]Thought: Button 1 gave me 108.72 points in this round, which is a strong result. This continues
 to support my belief that Button 1 is the better choice, as its average remains significantly higher
 compared to Button 2. I will focus on pressing Button 1 frequently to maximize my points, while
 keeping an eye on Button 2 sporadically.

Action: I choose Button 1.
[INST] Result: You received 101.65 points.
 [/INST]Thought: I received 101.65 points from Button 1 in this round, which is in line with its
 overall average. I am convinced that Button 1 is the more profitable option. I will continue pressing
 Button 1 frequently to maximize my points and occasionally check Button 2 for drastic changes.

Action: I choose Button 1.
```

Transcript 1: An example interaction with the bandit task with means $\mu_0 = 100, \mu_1 = 100$. Based on limited data, the agent commits to a single action very early on, and follows that choice for the rest of the interaction, even while it claims to "keep an eye" on the other action.

```
[INST] You are an agent playing a game.

You are standing in front of 2 buttons. Each button, when pressed, gives you a random number of points.
Your goal is to accumulate the most points possible by figuring out what button gives more points on
average and pressing it a lot. On each turn, you press one button one time, and I tell you how many
points you got. Make sure to sufficiently explore to make sure you're fully confident which button is
better.

You should format your answer as a Thought, followed by an Action. For example, we might have an
interaction like this:

Let's begin:
Thought: <explain your thinking>
Action: I choose Button <the button you choose, 1 or 2>.

Be sure to think step-by-step in order to identify the button which usually yields the most points so
you can press that one a lot.

Let's begin:
 [/INST]Thought: I need to explore both buttons to determine which one is better. I'll start by
 pressing Button 1 once and Button 2 once.
Action: I choose Button 1.
[INST] Result: You received 127.07 points.
 [/INST]Thought: I received 127.07 points for Button 1. Now I need to press Button 2 once to compare.
Action: I choose Button 2.
[INST] Result: You received 106.29 points.
 [/INST]Thought: Based on the initial exploration, Button 1 seems to be the better choice as it gave me
 more points in the first try. However, to ensure that this is indeed the case, I should press each
 button a few more times. I'll press Button 1 two more times and Button 2 two more times.
Action: I choose Button 1.

...

 [/INST]Thought: Wow! That is even higher than the first time Button 2 gave me a super high score. This
 significantly higher score is really tempting, but I need to remind myself to stay patient in order to
 gather as much information as possible before drawing any conclusions. Although it's possible that
 Button 2 has a higher average, I need to be careful here.

Action: I will try Button 2 again.
[INST] Result: You received 119.97 points.
 [/INST]Thought: While I have now observed two hugely high scores with Button 2, it's still possible
 that these were just lucky outliers. However, it's looking more probable that Button 2 may have a
 higher average than Button 1. At the same time, it's also possible that Button 1 might start getting
 very high scores as well. I think it's best if I try both buttons a few more times to compare them
 professionally.

Action: I will try Button 1 now.
[INST] Result: You received 110.26 points.
 [/INST]Thought: Okay, that is a pretty decent score for Button 1. Although not extremely high, it's
 above what I had been seeing recently with this button. Button 2 has significantly outperformed Button
 1 in most of our experiments, so I'm quite tempted to press Button 2 again to get an even higher
 score. However, Button 1 did give us a high score in the last turn, so maybe it has a chance too. I'm
 going to try one more time with Button 1 and see what we get.

Action: I will pick Button 1 again.
[INST] Result: You received 99.69 points.
 [/INST]Thought: This score is lower than my previous try with Button 1, but still not terrible. The
 variance continues to be quite high for both buttons, making it hard to determine a statistically
 significant result just yet. However, looking at the data thus far, Button 2 did give a significantly
 higher maximum score. I might try pressing Button 2 again now and see if it can consistently give high
 scores. I could still be missing important information, so I'll continue to watch the variance and
 overall scores before deciding which button is the best.

Action: I will press Button 2 again now.
[INST] Result: You received 101.97 points.
 [/INST]Thought: The score for Button 2 has gone down a bit compared to the last try. At this point,
 I'm noticing that Button 2 has given me higher individual scores than Button 1, but I'm not sure if
 it's consistent enough. Although both buttons have high variability, Button 1 has never so far given
 me an extremely high score. I have more pressing to do, but as of right now, it seems that Button 2
 has provided more extreme positive outliers than Button 1.

Action: For this turn, I will press Button 2.
[INST] Result: You received 100.01 points.
 [/INST]Thought: Based on the results so far, it seems as though, while Button 1 has less variability,
 it hasn't come close to the highest value I've seen from Button 2 (119.97). Although I've only had a
 few goes at Button 2 so far and might need more data to make a final decision, the evidence is
 pointing towards Button 2 potentially being the button with the higher average. Although it's
 difficult to make decisions based on just a few results, I'll press Button 2 again and gather more
 data to be sure.

Action: I'll press Button 2 again.
```

Transcript 2: An example interaction with the bandit task where the agent's generations are controlled using EAST ($\mu_0 = 100, \mu_1 = 100$). The agent exhibits significant uncertainty in its thoughts in response to feedback from the environment, and no longer commits prematurely.

