# OpenReview forum: "Controlling Large Language Model Agents with Entropic Activation Steering"
_ICLR.cc/2025/Conference — ICLR 2025 Conference Withdrawn Submission_

### Official Review · Reviewer_3b1k · 2024-11-02

**Soundness:** 3
**Presentation:** 3
**Contribution:** 3
**Rating:** 6
**Confidence:** 4

**Summary:**

This paper addresses the problem of how to encourage and control the exploratory behaviors of LLM agents in decision-making settings and derives a method from activation steering in LLM representation spaces.

The proposed method first extracts a steering vector from a dataset of interactions, which can lead to high action entropy. Then apply the steering vector during LLM's autoregressive unrolling by adding it to the activations at each position of generated tokens to modify the LLM agent’s behavior.

Results and analysis on several variants of two-armed bandit problems demonstrate the effectivess of the proposed method to modulate the uncertainty in the LLM’s thoughts and the ability to guide an LLM agent towards more exploratory actions, in contrast to token-level temperature sampling.

**Strengths:**

1. The paper presents and addresses a fundamental problem considering the uncertainty and explorative behaviors of LLM agents, since an LLM is typically pretrained or fine-tuned via autoregressive modeling from offline datasets, instead of an online setting where exploration can be done and uncertainty can be managed through the interaction between an agent and environment.
2. The proposed method is well derived and analyzed from a representation-level perspective, which is similar to recent work on activation steering for LLM generation, but well defined in decision-making settings.

**Weaknesses:**

1. The effectivess of proposed method, EAST, has only been validated in simple two-armed bandit problems, despite they are traditional tasks to investgate the exploratory capacity of agents .
Essentially, two-armed bandit problems are not sequential decision-making tasks. And to demonstrate the effectiveness of EAST in sequential decision-making settings, it is suggested to consider more sophiscated tasks , such as text-based games (e.g., ALFWorld[1]) or robotics/embodied tasks (e.g., Habitat[2]), to test how EAST might handle longer-term dependencies or more complex state spaces in these environments
2. The layout of Section 6 is not reader-friendly. A different organization of figures would enhance readability.
3. In Section 6.3, it is demonstrated that EAST can generalize to different prompt variations, however, it is still the same task. How EAST might perform when transferring between tasks with different reward structures or state spaces would be expected.

If all the concerns can be addressed well, I will consider to raise my score.

References:

[1] Shridhar, M., Yuan, X., Côté, M., Bisk, Y., Trischler, A., & Hausknecht, M.J. (2020). ALFWorld: Aligning Text and Embodied Environments for Interactive Learning. ArXiv, abs/2010.03768.

[2] Szot, Andrew et al. “Habitat 2.0: Training Home Assistants to Rearrange their Habitat.” ArXiv abs/2106.14405 (2021).

**Questions:**

1. Except for the differences in applying activation steering between an LLM agent and a natural LLM as stated in the paper, what are the key challenges when applying activation steering to modify an LLM agent's behavior, compared to using activation steering to modify an LLM's token generation?

---

### Official Review · Reviewer_a7z2 · 2024-11-03

**Soundness:** 2
**Presentation:** 3
**Contribution:** 2
**Rating:** 5
**Confidence:** 4

**Summary:**

This paper introduces a novel activation steering method aimed at reducing overconfident (less exploratory) behaviour in LLM-based dialogue agents. Experiments using Mixtral-8x7B and DBRX models demonstrate the effectiveness of this approach in a two-arm bandit setting.

**Strengths:**

- the paper is well-written and structured.
- the analysis of linking activation steering to more interpretable LLM behaviour is compelling and the insights are interesting.

**Weaknesses:**

- the motivation for increased uncertainty in LLM agent behaviour is ambiguous. Situations, where this exploratory behaviour would be beneficial, are not clearly defined, especially since uncertainty often correlates with less reliable and potentially unsafe outcomes.  Given this and the known variability of LLM outputs under fixed sampling temperatures, it is difficult to assess the practical value of this work.

- readers may also wonder about real-world applications of more exploratory LLM behaviour, as the paper lacks concrete examples of where such behaviour would be advantageous.

- the two-arm bandit environment used for evaluation limits the depth of exploration observed in the agents. This simple setting raises questions about how well the findings might transfer to environments with high-dimensional action spaces, which might showcase or potentially weaken the support for the authors' claims.

- additional experiments using larger, state-of-the-art (SOTA) models in Sections 4.1 and 6.1 could strengthen the study, as SOTA models are likely fine-tuned with more preference data to mitigate overconfidence effectively.

- the study’s generalizability claims are not well-supported by the experimental setup. Minor variations in task prompts are unlikely to yield substantial changes in LLM outputs, making these tests insufficient to support robust claims about generalizability. Expanding the tests to cover more varied task types would strengthen reliability.

- the colour scheme in Figure 1 is slightly confusing; for example, the role of the purple circle is unclear.


- missing appendix indexing reduces readability, e.g., lines 180, 375, etc.

**Questions:**

1. In Figure 1, do the activations correspond to different layers of the LLM or to the same layer $l$?

2. Could the authors provide more details on how the "shuffled" vector in Section 6.2 was constructed, particularly how it changes the direction of the steering vector?

3. Could the authors explain why, in Figure 9, the randomized vector is more resistant to multiplier beta concerning the fraction of valid completions? Also, it would be interesting to include a visualization of the calculated steering vector.

4. Have the authors considered using multiple steering vectors for varied interaction datasets, as a single vector may be insufficient to control LLM behaviour in complex environments? any discussion is appreciated.

---

### Official Review · Reviewer_bCJb · 2024-11-03

**Soundness:** 2
**Presentation:** 2
**Contribution:** 1
**Rating:** 3
**Confidence:** 4

**Summary:**

The authors propose a method to control the exploration of LLM in sequential decision-making tasks.
They experimentally show that in a 2 arms bandit setup, the entropy over actions of LLM significantly decreases the more decisions are taken (leading to insufficient exploration).
The method increases the entropy by steering the activations of the LLM.
The steering vector is computed a priori by doing forward pass in the network and putting more weight on directions that increases the entropy.

**Strengths:**

- the paper proposes a new way to compute steering vector to increase the entropy in 2-arms bandit problems
- it is clear enough and structured

**Weaknesses:**

- the paper lacks a clear formalism, what is the objective function we aim to optimize? what would the optimal solution look like? what are the difference between the optimal solution and the proposed one? why is it easier to obtain than the optimal solution? etc. those questions are not studied
- the choice of the baselines (scaling temperature) is limited. The proposed approach should for instance be compared with [1] (Equation 4).
- the generalization experiment is a bit limited, it is not clear that we could use the same steering vector to improve exploration in more general sequential decision problems (like for instance ALFWorld or BabyAI text)

[1] Liu, Sheng, et al. "In-context vectors: Making in context learning more effective and controllable through latent space steering." arXiv preprint arXiv:2311.06668 (2023).

**Questions:**

- Why Figure 6 does not contain the entropy comparison as in Figure 2?
- if you compute the steering vector on the 2 arms bandit problem, does it generalize to higher number of arms?

---

### Official Review · Reviewer_DQ96 · 2024-11-03

**Soundness:** 2
**Presentation:** 3
**Contribution:** 2
**Rating:** 5
**Confidence:** 3

**Summary:**

This paper investigates the use of steering activation vector to control the exploration of an LLM agent, represented as a higher degree of uncertainty and resulting in a more diverse high-level action distribution. The computation of steering vector involves estimating the resulting entropy of a prompt and using the entropy as weights to combine diverse latent activations. The resulting steering vector can then be added to the latent activation during inference to control the uncertainty in the LLM agent's thoughts. The proposed technique is evaluated under a simple environments where the agent should choose a better button out of two buttons.

**Strengths:**

1. This work seems to be the first application of activation steering to improve the diversity of decisions of LLM agents in sequential decision-making scenarios.
2. The proposed technique is able to effectively increase the entropy of the action distribution of the LLM policy, which could not be achieved by tuning the sampling temperature.
3. The authors show that the steering vector effectively introduces uncertainty during the reasoning process of the LLM agent.

**Weaknesses:**

1. The experiments are performed in relatively simple scenarios. Including more complex tasks, such as mathematical reasoning tasks GSM or MATH, can significantly improve the reliability of the proposed technique.
2. An ablation study about the entropy weight and normalization in Eq. (1) is lacked.

**Questions:**

1. Why is the steering vector computed from embedding of the last tokens of prompts but added to all tokens during the inference time?
2. In Fig 6., while EAST improves the uncertainty, the accuracy also decreases with EAST. Does there exist a tradeoff between exploration and exploitation when adopting EAST for controlling the exploration behavior of LLM agents?

---

### Note · Authors · 2024-11-22

**Comment:**

Due to external factors, we are withdrawing the paper from submission. We thank the reviewers for their time, attention, and constructive feedback on our work.

**Withdrawal Confirmation:**

I have read and agree with the venue's withdrawal policy on behalf of myself and my co-authors.